# Patient-Derived Tumoroid for the Prediction of Radiotherapy and Chemotherapy Responses in Non-Small-Cell Lung Cancer

**DOI:** 10.3390/biomedicines11071824

**Published:** 2023-06-26

**Authors:** Anasse Nounsi, Joseph Seitlinger, Charlotte Ponté, Julien Demiselle, Ysia Idoux-Gillet, Erwan Pencreach, Michèle Beau-Faller, Véronique Lindner, Jean-Marc Balloul, Eric Quemeneur, Hélène Burckel, Georges Noël, Anne Olland, Florence Fioretti, Pierre-Emmanuel Falcoz, Nadia Benkirane-Jessel, Guoqiang Hua

**Affiliations:** 1Regenerative Nanomedicine Unit, Center of Research on Biomedicines of Strasbourg (CRBS), French National Institute of Health and Medical Research (INSERM), University of Strasbourg, UMR 1260, 1 Rue Eugène Boeckel, 67000 Strasbourg, France; anas-2020@hotmail.fr (A.N.); jo.seitlinger@gmail.com (J.S.); charlotte.ponte@chru-strasbourg.fr (C.P.); julien.demiselle@chru-strasbourg.fr (J.D.); yidouxgillet@unistra.fr (Y.I.-G.); veronique.lindner@chru-strasbourg.fr (V.L.); anne.olland@chru-strasbourg.fr (A.O.); fiorettioce@gmail.com (F.F.); pefalcoz@gmail.com (P.-E.F.); 2Faculty of Dental Surgery, Strasbourg University Hospital (HUS), University of Strasbourg, 8 Rue de Ste. Elisabeth, 67000 Strasbourg, France; 3Department of Medical Intensive Care, Strasbourg University Hospital (HUS), 1 Place de l’Hôpital, 67000 Strasbourg, France; 4Department of Biochemistry and Molecular Biology, Strasbourg University Hospital (HUS), 67098 Strasbourg, France; erwan.pencreach@chru-strasbourg.fr (E.P.); michele.beau@chru-strasbourg.fr (M.B.-F.); 5Department of Pathology, Strasbourg University Hospital (HUS), 1 Place de l’Hôpital, 67000 Strasbourg, France; 6Transgene SA, 400 Boulevard Gonthier d’Andernach—Parc d’Innovation—CS80166, 67405 Illkirch Graffenstaden, France; balloul@transgene.fr (J.-M.B.); quemeneur@transgene.fr (E.Q.); 7Department of Radiation Oncology, Institut de Cancérologie Strasbourg Europe (ICANS), UNICANCER, 67200 Strasbourg, France; h.burckel@icans.eu (H.B.); g.noel@icans.eu (G.N.); 8Radiobiology Laboratory, Paul Strauss Comprehensive Cancer Center, Institut de Cancérologie Strasbourg Europe (ICANS), 17 Rue Albert Calmette, 67033 Strasbourg, France; 9ICube Laboratory, 300 Bd Sébastien Brant, 67400 Illkirch-Graffenstaden, France; 10Lung Transplantation Group, Thoracic Surgery Department, Strasbourg University Hospital (HUS), 1 Place de l’Hôpital, 67000 Strasbourg, France

**Keywords:** patient-derived tumoroid, radiation therapy, chemotherapy, personalized medicine, non-small-cell lung cancer

## Abstract

Radiation therapy and platinum-based chemotherapy are common treatments for lung cancer patients. Several factors are considered for the low overall survival rate of lung cancer, such as the patient’s physical state and the complex heterogeneity of the tumor, which leads to resistance to the treatment. Consequently, precision medicines are needed for the patients to improve their survival and their quality of life. Until now, no patient-derived tumoroid model has been reported to predict the efficiency of radiation therapy in non-small-cell lung cancer. Using our patient-derived tumoroid model, we report that this model could be used to evaluate the efficiency of radiation therapy and cisplatin-based chemotherapy in non-small-cell lung cancer. In addition, these results can be correlated to clinical outcomes of patients, indicating that this patient-derived tumoroid model can predict the response to radiotherapy and chemotherapy in non-small-cell lung cancer.

## 1. Introduction

Lung cancer is the worldwide leading cause of death; almost 25% of all cancer deaths are due to this disease in the US [1]. In 2020, more than 2 million people in the world were diagnosed with lung cancer. Early diagnoses and complete resection surgeries would be helpful for non-small-cell lung cancer (NSCLC) patients, which will increase their 5-year survival rate up to 40%. Unfortunately, most patients are asymptomatic in the early stage of lung cancer, and as a result, most of them are found to have this poor prognosis of disease at an advanced/metastatic stage [2]. Consequently, such resection surgery cannot be carried out for patients at the advanced and metastatic stages [3]. As a result, the general five-year survival rate of NSCLC remains low [4]. 

It was reported that more than 70% of lung cancer patients could have an evidence-based indication for radiotherapy, which can be used to reduce the tumor size before lung cancer surgery or to eradicate any remaining cancer cells left in the lungs after surgery [5]. However, one recent phase 3 clinical trial suggested that postoperative radiotherapy cannot be recommended as the standard of care in patients with stage IIIAN2 NSCLC [6]. Platinum-based chemotherapy regimens are administered to patients as a gold-standard treatment, but remain limited by severe and dose-limiting side effects [7]. Recent therapies targeting specific genomic mutations such as EGFR, KRAS, ALK, RET, ROS1, BRAF and ERBB2, as well as immunotherapies, could provide a better survival outcome for patients who have a somatically activated oncogene in their tumors [8,9,10,11,12,13] or pre-existing immunity against cancer [14]. However, these promising therapies are rapidly compromised by resistance due to the heterogeneity of the tumor or low complete response rate to immunotherapy [15,16,17,18]. In this context, a personalized pre-clinical drug-screening system that can mimic the real in vivo clinical situation is urgently required for NSCLC patients not only to recommend an efficient drug treatment with lower toxicities and to improve the overall survival and quality of life but also to reduce the increasing medico-economic impact of anti-cancer therapies. 

We and other researchers have demonstrated that patient-derived organoid/tumoroid (PDT) models maintain both histopathological and genomic characterization of parental tumors in lung cancer [19,20,21,22,23], which could consequently be used to predict the responses of treatments. The lung cancer organoid model was shown to predict the response to cytotoxic drugs, as well as targeted treatment [20]. Delom et al. have found that 3D tumoroids were more resistant to radiation than 2D monolayer cultures [22]. However, no research on lung cancer has demonstrated that the PDT model is feasible to predict the radiation therapy response. 

We have recently published a PDT model for NSCLC patients that could allow us to propose the most efficient back-to-patient treatment within two weeks [23]. Here, we further showed that different dose–response profiles were indeed observed using our PDT model for X-ray-based radiotherapy and cisplatin/vinorelbine-based chemotherapy. Combing the experimental results from these PDTs with the clinical outcomes from patients, we demonstrate here that the PDT model established in our lab could indeed provide a predictive indication for clinical therapies proposed to patients.

## 2. Materials and Methods

### 2.1. Human Specimens, Tissue Preparation and PDT Formation

The research project was conducted in accordance with the Declaration of Helsinki and approved by the Ethics Committee of Grand Est, France (CNRIPH N° 20.11.12.42058). Samples are taken from patients who underwent lung major resection for localized lung cancer in the Thoracic Surgery Department of Strasbourg University Hospital. All written informed consent was obtained from patients the day before the surgery. As soon as the lung resection was performed, an adenocarcinomas sample (5 mm by 5 mm fragment) was freshly taken by a pathologist without interfering with the clinical pathological diagnosis and then immediately transported in cold culture medium to the laboratory for the formation of PDTs. The tumor dissociation and PDT formation were carried out as previously described [23]. A total of 5000 primary human pulmonary fibroblasts (HPF, CP3300-SC, (CliniSciences, Nanterre, France)) were mixed with 5000 patient-derived cells from either tumor sample in 150 µL of mixed medium (50% of HPF medium + 50% of DMEM/F12 supplemented with 10% FCS, 20 ng/µL bFGF, 50 ng/µL human EGF, 2% B27 and 1% N2) in each well of a 96-well round bottom ultra-low-attachment (ULA) plate (S-bio, Fuggerstraße, Germany). 

### 2.2. X-ray Irradiation

The X-ray irradiation was performed in the Centre Paul Strauss (ICANS, Strasbourg, France) with a 6 MV linear accelerator (Siemens, Courbevoie, France). The PDTs were irradiated directly in the 96-well ultra-low-attachment (ULA) plate after 4 days of formation with an either 4 Gy or 8 Gy X-ray dose at a dose rate of 2 Gy/min with a field size of 40 × 40 cm, a gantry angle of 180° and a source isocenter distance of 100 cm. The PDTs were then incubated again at 37 °C with 5% CO_2_. γ-H2AX staining was performed 24 h post-irradiation. A cell viability assay and cleaved-Caspase 3 staining were performed 4 days post-irradiation.

### 2.3. Chemotherapy Treatment

The PDTs were treated after 4 days of formation with different concentrations of cisplatin for 96 h. The medium with the drug was changed after 48 h. For the cisplatin and vinorelbine co-treatment, 10µM of vinorelbine was added with each concentration of cisplatin [24]. A total of 8 PDTs were treated in each condition. 

### 2.4. Cell Viability Assay

After the treatment, the cell viability of the PDTs was assessed with a CellTiter-Glo-3D Cell Viability Assay (Promega, Madison, WI, USA). Each PDT contained in 40 µL of the medium in which they had been incubated was transferred from ULA plate to an opaque-walled 96-well plate. A 40 µL volume of CellTiter-Glo 3D reagent (Promega, Madison, WI, USA) were added to each well, and the plate was agitated for 25 min at room temperature. The luminescence resulting from the release of ATP was read by MicroBeta TriLux Luminescence Counter (PerkinElmer, Waltham, MA, USA). Each luminescence value was normalized to the mean value of the control (PDTs without treatment), which is predefined as 100%.

### 2.5. Histological and Immunofluorescence Staining

PDTs were fixed in Tissue-Tek^®^ OCT (Optimum Cutting Temperature, Fisher Scientific, Illkirch, France) and frozen at −20 °C. Samples in 10 μm sections were cut with a Leica CM3000 cryostat (Leica Biosystems, Nanterre, France) and fixed in 4% paraformaldehyde (PFA) for 10 min at room temperature (RT). The standard hematoxylin and eosin staining protocol was followed.

For the cleaved-Caspase 3 staining, PDT sections were fixed with PFA, washed three times with PBS and then incubated in PBS containing 1% BSA and 0.1% Triton X-100 for 30 min at RT. After the wash, the sections were incubated with cleaved Caspase 3 (D175, Cell Signaling Technology, Danvers, MA, USA) overnight at 4 °C. The primary antibody was detected by incubating with Alexa Fluor™ 488 (A11001, Invitrogen, Illkirch, France) secondary antibody for 1 h at RT. The samples were washed with PBS before incubation with 200 nM DAPI (Sigma-Aldrich, Saint-Quentin-Fallavier, France) for 10 min at RT. The slides were observed under an epifluorescence Leica DM4000 B microscope (Leica Biosystems, Nanterre, France).

For the whole-mount staining of γ-H2AX, the PDTs were fixed with PFA at 4 °C overnight (ON), washed three times with PBS with rotation for 15 min and then incubated in PBS containing 1% BSA and 0.1% Triton X-100 with rotation at RT ON. After 3 washes with rotation, the PDTs were incubated with γ-H2AX antibody (clone JBW301, 05-636, Sigma-Aldrich, Saint-Quentin-Fallavier, France) at 4 °C with rotation ON. Incubation with Alexa Fluor™ 488 (A11001, Invitrogen, Illkirch, France) secondary antibody was also carried out with rotation at RT ON. The samples were washed with PBS with rotation before incubation with 200 nM DAPI (Sigma-Aldrich, Saint-Quentin-Fallavier, France) at RT ON with rotation. Images were taken and analyzed with a confocal microscope (Leica SP8x, objective HC FLUOTAR 25X/0.95, Leica Biosystems, Nanterre, France). 

### 2.6. Quantitative PCR Analyses

The RNA of 10 PDTs of each treatment were extracted with Direct-zol RNA MicroPrep (ZYMO research, R2062, Freiburg im Breisgau, Germany) according to the manufacturer’s protocol. The RNA concentrations and purity were measured using the NanoDrop ND-1000 spectrophotometer (NanoDrop Technologies, Rockland, DE, USA). Reverse transcription was conducted with the iScript^®^ reverse Transcription Supermix (Bio-Rad, Marnes-la-Coquette, France). The quantitative PCR reaction was then performed using the iTaq^®^ Universal SYBR^®^ green super mix (Bio-Rad, Miltry-Mory, France) using CFX Connect-Real-Time PCR Detection System (Bio-Rad, Miltry-Mory, France). The primers were synthesized by Life Technologies (Saint-Aubin, France) (Table 1). The specificity of the reaction was controlled using melting-curve analysis. GAPDH was used as the endogenous RNA control (housekeeping gene). The expression level was calculated using the comparative Ct method (2^ΔΔCt^) after normalization to GAPDH. All PCR assays were performed in triplicate, and the results are represented by the mean values with standard error bars.

### 2.7. Droplet Digital PCR

A mutation analysis was performed using duplex ddPCR mutation assays from BioRad on a QX200 Droplet Digital PCR system (Marnes-la-Coquette, France). All reactions were prepared using the ddPCR Supermix for Probes (Bio-Rad, Marnes-la-Coquette, France) and performed in duplicate. KRAS somatic mutations were detected using commercial probes provided by Bio-Rad (Marnes-la-Coquette, France). For each mutation, two different labeled probes are tested in a single reaction, the first one to detect the mutant allele (6-carboxy-fluorescein, FAM) and the second one to detect the wild-type allele (Hexachloro-fluorescein, HEX). The quantitative value of the KRAS mutant was determined using the variant allele frequency (VAF%). The VAF was reported by the QX200 Droplet Reader and QX manager software, version 2.1 (Bio-Rad, Marnes-la-Coquette, France), after correction for the Poisson distribution.

## 3. Results

### 3.1. Identification of Different Dose–Responses of Lung Cancer PDTs to X-ray-Based Radiation Therapy

Non-small-cell lung cancer samples were taken from patients who underwent major lung resection for localized lung cancer in the Thoracic Surgery Department of Strasbourg University Hospital. The general information, pathological characterization and clinical outcome of the patients are summarized in Table 2.

NSCLC samples freshly resected by thoracic surgeons were immediately transported in cold culture medium to the laboratory for the formation of patient-derived tumoroids (PDTs) as previously described [23]. At day four, the PDTs were irradiated with X-rays at 4 Gy in one fraction, and cell viability assays and immunofluorescence analyses were performed 96 h post-irradiation. As shown in Figure 1A, different dose–response profiles were observed for PDTs formed from 11 patients. Around 50% of cells were found dead in the PDTs formed from patients 21T080, 211T117 and 21T135, whereas few dead cells were detected in the PDTs formed from patients 20T222, 21T099, 21T324, 21T362 and 21T518. These results suggest that patients 21T080, 211T117 and 21T135 could be sensitive to X-ray-based radiotherapy; however, patients 20T222, 21T099, 21T324, 21T362 and 21T518 may present radioresistance. The immunofluorescence assay revealed that cleaved-Caspase 3 expression was indeed observed in the PDTs formed from the patient with a good radiation-therapy-related response (21T117), which demonstrated the presence of apoptotic cells in the PDTs. No apoptotic cells were labeled in PDTs formed from the patient with no significant response in viability (21T324) (Figure 1B).

It is known that radiation exposure causes DNA double-strand breaks (DSBs), which subsequently lead to cell death or cancer formation, and DSBs could be identified and quantified with the immunofluorescence assay using the γ-H2AX antibody, which could consequently reflect the radiosensitivity of the cells or PDTs [25]. The PDTs were fixed 24 hr post-irradiation to investigate the residual remaining DSBs. Notably, γ-H2AX foci were stained inside of the nucleus upon irradiation only in PDTs formed from patient 21T117, but not in those formed from 21T324, indicating that fewer DNA lesions made by irradiation were detected 24 h post-irradiation in patient 21T324 than in patient 21T117 (Figure 2A). These results suggest that X-ray-based radiotherapy may be beneficial for patient 21T117, but not for patient 21T324.

Regarding radiation therapy administration, depending on the clinical indications, patients receive the total amount of irradiation dose with repetitive fractions (up to 25 or 30 fractions depending on the regimen) within a short period of time (3 to 6 weeks mainly). Thus, we have irradiated repetitively those radiosensitive-profile PDTs once every two days to evaluate the DNA DSBs by quantifying the number of stained γ-H2AX foci in the nucleus. The PDTs were fixed 24 h after the last irradiation and whole-mount immunofluorescence staining of γ-H2AX was performed. Quantitative analyses were performed with images obtained from a confocal microscope. We assessed the percentage of γ-H2AX-positive cells per layer (20 layers, 5 µm depth) and the number of stained γ-H2AX foci in each positive cell (100 cell nuclei) in each PDT. Both the percentage of γ-H2AX-positive cells per layer and the number of γ-H2AX foci in each cell nucleues were significantly increased by the repetitive irradiation (Figure 2B). We have also observed that 8 Gy irradiation effected more DSBs than 4 Gy irradiation in radiosensitive PDTs (Figure 1A).

### 3.2. Identification of Different Dose–Responses of Lung Cancer PDTs to Cisplatin-Based Chemotherapy

To validate our PDT model for chemotherapy, PDTs formed from ten patients were treated at day four with cisplatin. Three distinct groups were identified with cell viability assays after 4 days of treatment (Figure 3A). Resistance to cisplatin treatment was observed in the PDTs formed from patients 21T226, 21T201 and 21T176 (IC_50_ > 30 µM). Good dose–response curves were obtained in the PDTs formed from patients 21T518, 21T581, 20T222, 21T080 and 21T362, while the PDTs formed from patients 20T428 and 22T017 were very sensitive to cisplatin treatment (IC_50_ < 7 µM). H&E staining revealed that the PDT structures and size were gradually destroyed and decreased, respectively, with an increasing dose of cisplatin in the PDTs formed from patients 21T080 and 21T581 (Figure 3B). Not surprisingly, the structure of the PDTs formed from patient 21T017 was damaged at 5µM, while the PDTs formed from patient 21T226 still maintained their structure even with 20 µM of cisplatin (Figure 3B). Quantitative PCR analyses were performed on the PDTs to investigate the expression of the pro-inflammatory genes Caspase 3 and Bcl-2–associated X protein (BAX). As shown in Figure 3C, the expression of these two genes increased with cisplatin treatment. However, compared to that of 21T362, a weaker expression of Caspase 3 and BAX was observed in the PDTs formed from patient 21T201, in which a resistance to treatment was detected. 

In clinical chemotherapy treatment for lung cancer, cisplatin (or carboplatine) is administrated to patients with another drug, such as vinorelbine, pemetrexed or paclitaxel (double platinum-based chemotherapy). Thus, PDTs were treated with the combination of cisplatin and vinorelbine. No significant effect was observed with vinorelbine alone (10 µM); however, compared to the treatment with cisplatin alone, the combined treatment decreased the IC_50_ of the PDTs formed from patients 21T226 (IC_50_: 26.1 µM), 21T080 (IC_50_: 11 µM) and 21T581 (IC_50_: 6.5 µM), indicating that the combined cisplatin/vinorelbine treatment provided a better efficiency (Figure 3D). Interestingly, a resistance to the combined chemotherapy was also observed in the PDTs formed from patient 20T222 (IC_50_: 17.6 µM), in which the treatment with cisplatin alone gave a better dose response.

### 3.3. Correlation of Bench Results from PDT Models with Clinical Outcome Assessment

Since different “sensitive–resistant” profiles were observed in patients upon either radiation therapy or chemotherapy, which are commonly administrated to NSCLC patients in clinics, we investigated whether there is a correlation between the results obtained from PDT models and the clinical outcome assessments from the patients. Up to two years of follow-up was pursued for all the patients included in this study to evaluate the clinical outcome of the treatments decided after the tumor board meeting. 

According to these data, adjuvant microscopic margins involvement radiation was proposed for patient 21T117 upon pneumonectomy in March 2021 because of the observation of hilar lymph node invasion. Although there was no macroscopic target after surgery to assess the response to adjuvant radiation therapy in this patient, no cancer relapse despite the advanced stage was detected in February 2023, which indirectly suggests a good response to radiation treatment. In agreement with the clinical outcome of this patient, the PDTs formed from this patient showed a good response to radiation (Figure 1 and Figure 2). For patient 21T324, who had a lobectomy in June 2021, a lymph node recurrence/progression was observed and pathologically confirmed in February 2022. Since then, stereotactic body radiation therapy (SBRT) was performed on this patient. Our PDT model indicated that tumor cells from this patient presented radioresistance (Figure 1 and Figure 2), while the clinical results showed the size of the mediastinal lymphadenopathy was not reduced after SBBRT completion. Thus, it could be possible that the recurrent carcinoma cells found in the lymph nodes were resistant to radiation therapy. Patient 22T017, having had neoadjuvant chemotherapy before the lobectomy, had an encouraging response to platinum treatment. A very sensitive dose response was also revealed in our PDT model of this patient (Figure 3A,B). On the contrary, an increase in the tumor size was observed for patient 21T226, who received neoadjuvant cisplatin and vinorelbine chemotherapy before surgery, indicating that this patient is resistant to chemotherapy. In our PDT models, cisplatin and vinorelbine in combination (IC_50_: 26.1 µM) appeared to be more efficient than cisplatin alone (IC_50_: 30.4 µM); however, resistance to chemotherapy was observed in both treatments (Figure 3A,B,D). Patients 21T080 and 21T362, who had surgery in February 2021 and July 2021, respectively, both benefited from cisplatin and vinorelbine adjuvant chemotherapy, and no relapse has been detected until now for these two patients. A good dose response was observed in our PDT model for both patient 21T080 and patient 21T362 with cisplatin treatment, while an even more efficient treatment with cisplatin and vinorelbine chemotherapy occurred for patient 21T080.

## 4. Discussion

Patient-derived normal tissue or tumor organoid models have been reported to be able to predict the sensitivity of radiotherapy in several cancer types [26,27,28,29,30,31,32,33,34,35,36,37]; however, no PDT model had been demonstrated to predict the clinical outcome in lung cancer. In this study, we aimed to evaluate the potential benefit of the NSCLC PDT model previously published with X-ray-based radiation therapy treatment. Upon irradiation, DNA DSBs were observed in radiosensitive PDTs within 24 h (Figure 2A) and could be effectively enhanced either by an increased dose or by repetitive irradiation (Figure 2B). These DNA DSBs further led to cell apoptosis in 4 days (Figure 1A,B). On the contrary, DNA DSBs could not be well-detected in radioresistant PDTs, which did not permit us to conduct quantitative analyses; consequently, little cell death were found 96 h post-irradiation (Figure 1 and Figure 2). 

The KRAS mutation is detected in 30% of NSCLCs [38] and was reported to be associated with resistance to radiation therapy in NSCLC [39,40]. Among the 11 patients who were included in this study of radiation therapy of their PDTs, KRAS mutations were detected in needle biopsies of four patients (see Table 2). The PDTs formed from three of the four patients (20T222, 20T401 and 21T099) were analyzed with Droplet Digital PCR (ddPCR) to investigate the presence of KRAS mutations in the PDTs. Not surprisingly, the same KRAS mutations were detected in all PDTs: KRAS G12C in 20T222 (37.86% in PDTs vs. 4.7% in needle biopsy), KRAS G12V in 20T401 (33.9% in PDTs vs. 60% in needle biopsy) and KRAS G12D in 21T099 (6.14% in PDTs vs. 16% in needle biopsy). In agreement with these previous reports, resistance to radiation therapy was indeed observed in our PDT models. 

Different dose responses were observed in the PDTs after cisplatin-based chemotherapy. A KRAS G12C mutation was detected in five patients (see Table 2). Interestingly, among these five patients, resistance to cisplatin treatment was revealed in the PDTs formed from patients 21T226, 21T201 and 21T176, whereas a good dose response was observed to PDTs formed from other two patients (20T222 and 21T518) (Figure 3A). Clinical trials demonstrated that KRAS wild-type lung cancer patients exhibited better responses to chemotherapy than KRAS mutant patients; however, the difference was not significant [41,42,43]. Thus, KRAS status could not be used as an indicator to predict the efficiency of chemotherapy in lung cancer. In this study, although PDTs were formed only from a small cohort of ten patients, a significant dose–response profile could be distinguished to predict the efficiency of chemotherapy in NSCLC.

The resistance or sensitivity to radiotherapy or chemotherapy observed in our PDT models formed from other patients, who were not mentioned in the previous results and discussion, could not be concluded with their clinical outcomes, because these patients did not receive either radiotherapy or chemotherapy before or after the surgery. Although prospective trials evaluating the response to adjuvant therapy with a surrogate such as PDTs are needed, our model appears to identify patients with different response profiles that appear to be related to the clinical response for many. Our model also has the advantage of versatility, as it allows for the evaluation of the response to chemotherapy and also to radiotherapy. Our model can also evaluate the response to tyrosine kinase inhibitors (upcoming publication) and may allow the evaluation of immunotherapy treatment by considering the tumor microenvironment in the future. 

One limitation of our PDT model is that these PDTs were formed with tumor samples from the surgery, but not from a lung needle biopsy. It obviously made more sense to start with surgical specimens for feasibility reasons, but this limits the transferability of our model to advanced-stage lung cancer patients. Further work using needle biopsies will be required and would allow us to assess the response to treatment in neoadjuvant therapy settings. 

In the era of personalized treatments, our PDT model is a valuable aid to therapeutic decision making. The clinical presentation as well as the molecular and histological evaluation of the tumor will allow us to decide the treatment panel to be tested on organoids. Our model will then allow us to decide on the best treatment from the first line of therapy, and if possible, improve the overall survival of our patients. This approach could be part of a prospective trial where the therapeutic decision made at the tumor board would be guided by our evaluation on PDTs.

## 5. Conclusions

Our patient-derived PDT model allows us to evaluate the response to different treatments offered for lung cancer (chemotherapy, radiation therapy). This robust and straightforward model already allows us to obtain different response profiles depending on the patient and the therapy evaluated. We believe that this type of tool will be a valuable aid in the therapeutic decision in the future of personalized lung cancer treatments.

## Figures and Tables

**Figure 1 biomedicines-11-01824-f001:**
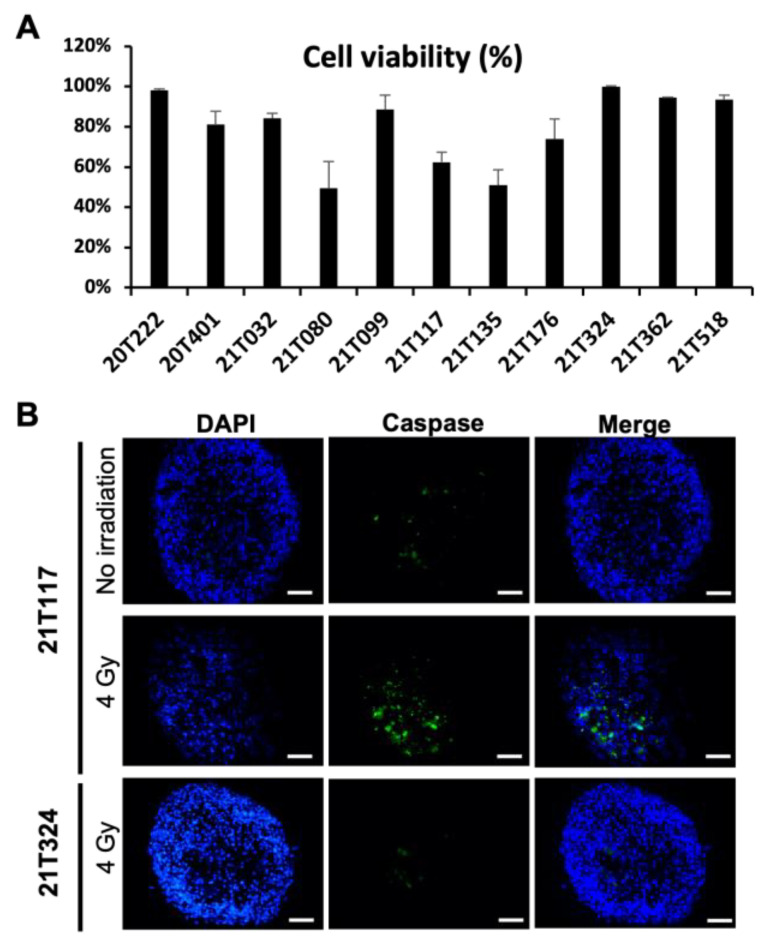
Validation of X-ray irradiation in PDT model. (**A**) Dose–response of PDTs formed from 11 patients’ tumor samples. (**B**) Immunofluorescence analyses performed on cryosection of PDTs with cleaved-Caspase 3 antibody. Scale bar: 100 µm.

**Figure 2 biomedicines-11-01824-f002:**
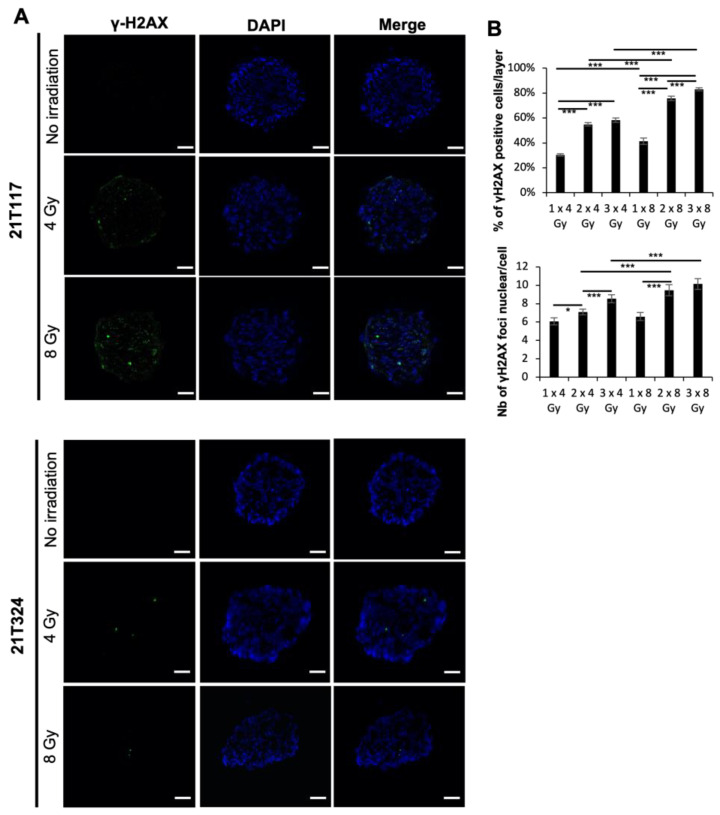
X-ray irradiation-induced DNA DSBs in PDTs. (**A**) Immunofluorescence analyses performed on cryosection of PDTs with γ-H2AX antibody. Scale bar: 100 µm. (**B**) Quantitative analyses of DNA DSBs in PDTs. *: *p* < 0.05; ***: *p* < 0.001.

**Figure 3 biomedicines-11-01824-f003:**
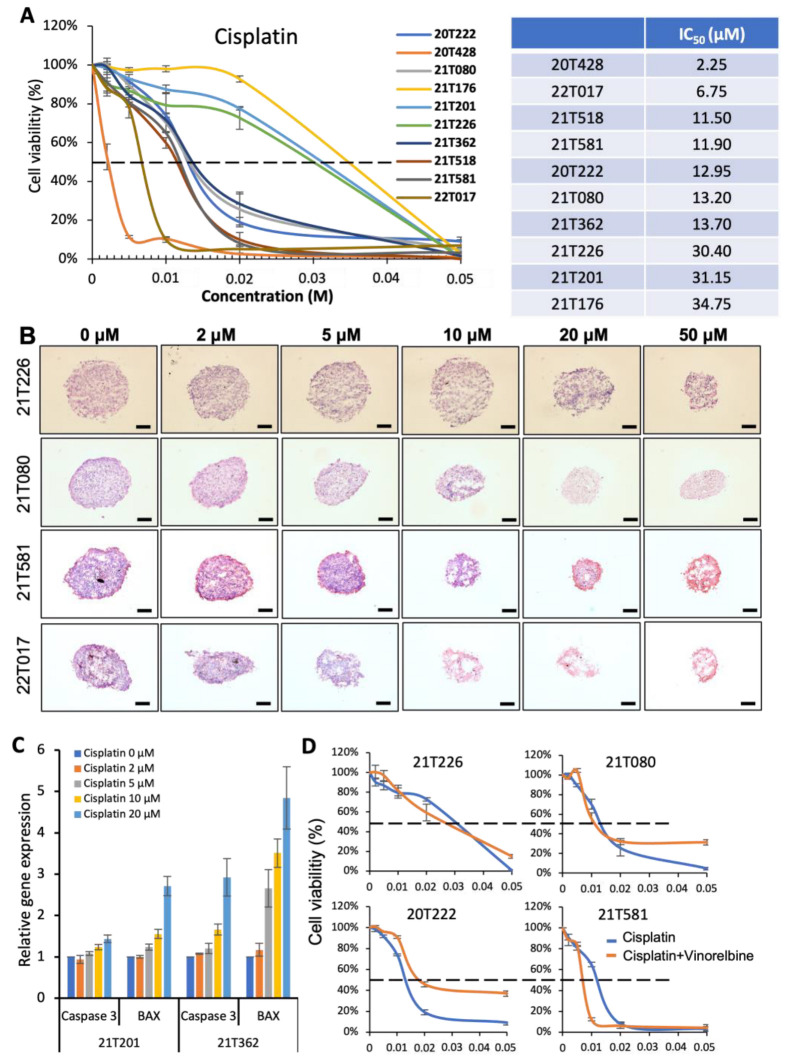
Validation of cisplatin-based chemotherapy in PDT model. (**A**) Dose–response of PDTs formed from 10 patients’ tumor samples with IC_50_. (**B**) Hematoxylin and eosin (H&E) staining performed on cryosection of PDTs 4 days post-treatment. Scale bar: 100 µm. (**C**) Quantitative PCR analyses on pro-apoptosis genes Caspase 3 and BAX. (**D**) Dose–response comparison between cisplatin and cisplatin + vinorelbine treatment in PDTs formed from four patients.

**Table 1 biomedicines-11-01824-t001:** List of primers used for quantitative PCR analysis.

Gene Product	Forward Primer Sequence	Reverse Primer Sequence
GAPDH	CTGACTTCAACAGCGACACC	GTGGTCCAGGGGTCTTACTC
Caspase 3	AGAACTGGACTGTGGCATTGAG	GCTTCTCGGCATACTGTTTCAG
BAX	GATGCGTCCACCAAGAAGCT	CGGCCCCAGTTGAAGTTG

**Table 2 biomedicines-11-01824-t002:** List of NSCLC patients’ samples used in this study for the generation of patient-derived tumoroids.

Patient Identity	Sourcing	Gender	Age	NSCLC Subtype and TNM Score	Mutational Status	Decision MDTMs
20T222	CRB Strasbourg	M	80	AC cT4N2Mx	KRAS G12C	Died quickly after surgery
20T401	CRB Strasbourg	F	64	AC cT3N0M0	KRAS G12V	Adjuvant chemotherapy
20T428	CRB Strasbourg	M	66	AC pT3N3M0	No mutation	Surveillance
21T032	CRB Strasbourg	M	77	SCC cT2aN0M0	Unknown	Surveillance
21T080	CRB Strasbourg	F	48	AC cT1cN0M0	EGFR deletion exon 19	Adjuvant chemotherapy
21T099	CRB Strasbourg	M	60	AC cT2bN0M0	KRAS G12D	Adjuvant chemotherapy
21T117	CRB Strasbourg	M	69	SCC cT3N0M0	Unknown	Adjuvant radiotherapy
21T135	CRB Strasbourg	M	71	AC cT1bN0M1a	No mutation	Adjuvant chemotherapy
21T176	CRB Strasbourg	F	55	AC cT1cN0M0	KRAS G12C	Surveillance
21T201	CRB Strasbourg	M	68	AC cT2aN1M0	KRAS G12C	Adjuvant chemotherapy
21T226	CRB Strasbourg	F	62	AC cT2aN2M0	KRAS G12C	Surveillance
21T324	CRB Strasbourg	M	69	AC cT2bN2M0	Unknown	Immunotherapy
21T362	CRB Strasbourg	M	65	AC cT2bN2M0	EGFR deletion exon 19	Adjuvant chemotherapy and radiotherapy
21T518	CRB Strasbourg	M	84	AC cT2bN0M1a	No mutation	No chemotherapy
21T581	CRB Strasbourg	M	63	AC cT1bN0M0	Unknown	Surveillance
22T017	CRB Strasbourg	M	79	AC cT2bN2M0	ROS 1	Adjuvant chemotherapy and radiotherapy

MDTM: multidisciplinary team meetings; AC: adenocarcinoma; SCC: squamous-cell lung carcinoma. TNM Classification of Malignant Tumors (TNM): T: size or direct extent of the primary tumor; N: degree of spread to regional lymph nodes; M: presence of distant metastasis.

## Data Availability

Not applicable.

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
