# Peer review of "Patient-Derived Tumoroid for the Prediction of Radiotherapy and Chemotherapy Responses in Non-Small-Cell Lung Cancer"

_biomedicines, 2023, doi:10.3390/biomedicines11071824_

Round 1
Reviewer 1 Report
The manuscript showed correlation between response of NSCLC PDT models to irradiation and chemotherapy and clinical response.
Since the sample size is small, the authors need to emphasize this weakness for the interpretation regarding the correlation. Although the data is interesting, there are some concerns:
1. The authors suggested the correlation between radiotherapy response observed in the PDT model and clinical response from 2 patients. One patient received pneumonectomy and the other lobectomy prior to radiotherapy. Since the surgery could play an important role here for the prevention of tumor relapse in patient 21T117, was there other criteria assessed to suggest good tumor response to radiotherapy in this patient eg. tumor shrinkage?
2. Please state rationale of using fixed concentration of Vinorelbine for the combination treatment
3. In the method section, there was no information regarding at which time point after the treatment was cleaved-caspase 3 analyzed by IF
4. Figure 1A: Quantitative data of cleaved-caspase 3 analyzed by IF will further strengthened conclusion
5. Figure2: Was the base line gammaH2AX different in the 2 PDT lines? Did the authors analyze gamma-H2AX at earlier time points? The conclusion about more rapid DSBs repair in the resistant line is not convincing without base line data. eg. It could be that there were less DSBs generated in the resistant cell line.
6. Figure 2: Quantitative data of gamma H2Ax the resistant line, 21T324, is important for comparison
7. Line 280: “even more efficient treatment with cisplatin and vinorelbine chemotherapy for 21T080” but there was no synergistic effect observed in figure 3D
8. How was the effect of single Vinorelbine treatment of the PDT models? Were there synergistic/ additive/antagonistic mechanisms of the 2 agents?
Author Response
Reviewer 1
The manuscript showed correlation between response of NSCLC PDT models to irradiation and chemotherapy and clinical response.
Since the sample size is small, the authors need to emphasize this weakness for the interpretation regarding the correlation. Although the data is interesting, there are some concerns:
- The authors suggested the correlation between radiotherapy response observed in the PDT model and clinical response from 2 patients. One patient received pneumonectomy and the other lobectomy prior to radiotherapy. Since the surgery could play an important role here for the prevention of tumor relapse in patient 21T117, was there other criteria assessed to suggest good tumor response to radiotherapy in this patient eg. tumor shrinkage?
We thank the reviewer for this excellent comment. These two patients underwent major lung resection (as pneumonectomy and lobectomy) followed by adjuvant radiation therapy. We should also acknowledge that these patients have been treated with adjuvant radiation therapy before the publication of the Lung ART trial by Cecile Le Pechoux in 2022 (Le Pechoux C, Pourel N, Barlesi F, et al. Postoperative radiotherapy versus no postoperative radiotherapy in patients with completely resected non-small-cell lung cancer and proven mediastinal N2 involvement (Lung ART): an open-label, randomised, phase 3 trial. Lancet Oncol. 2022;23(1):104-114. doi:10.1016/S1470-2045(21)00606-9.
We have modified the description of radiation therapy for the patient 21T117 in the second paragraph of the section 3.3. Correlation of bench results from PDT models to clinical outcome assessment: “adjuvant microscopic margins involvement radiation was proposed to the patient 21T117 upon pneumonectomy in March 2021 because of the observation of hilar lymph node invasion. Although there was no macroscopic target after surgery to assess the response to adjuvant radiation therapy in this patient, no cancer relapse despite advance stage was detected in February 2023 which indirectly suggest a good response to radiation treatment.”
- Please state rationale of using fixed concentration of Vinorelbine for the combination treatment
We thank the reviewer for this critical remark. Since till now, we cannot find any published data for the combined cisplatin/vinorelbine treatment in lung cancer organoid model. The fixed concentration of vinorelbine was determined based on other published studies. 10 µM of Vinorelbine was used in several research articles, such as Li Zhichao et al., Human Lung Adenocarcinoma-Derived Organoid Models for Drug Screening, iScience, Volume 23, Issue 8, 21 August 2020, 101411; Li Hongyang et al., DT-13 synergistically enhanced vinorelbine-mediated mitotic arrest through inhibition of FOXM1-BICD2 axis in non-small-cell lung cancer cells, Cell Death & Disease volume 8, pagee2810 (2017).
Now we have added a new reference 24 in the section 2.3. Chemotherapy treatment for the concentration of Vinorelbine.
- In the method section, there was no information regarding at which time point after the treatment was cleaved-caspase 3 analyzed by IF
We have added the information in the page 6, second paragraph of the section 3.1. Identification of different dose-response of lung cancer PDTs to X-ray based radiation therapy: “At day 4, PDTs were irradiated by X-ray at 4 Gy in one fraction, and cell viability assays and immunofluorescence analyses were performed 96-hr post-irradiation.”
- Figure 1A: Quantitative data of cleaved-caspase 3 analyzed by IF will further strengthened conclusion
We agree with the reviewer that cleaved-caspase 3 analyzed by IF will further strengthened the conclusion made from cell viability assay. That’s why we performed these two analyses at the same time point (96-hr post-irradiation) to demonstrate that more apoptotic cells were detected in radiosensitive PDTs.
- Figure2: Was the base line gammaH2AX different in the 2 PDT lines? Did the authors analyze gamma-H2AX at earlier time points? The conclusion about more rapid DSBs repair in the resistant line is not convincing without base line data. eg. It could be that there were less DSBs generated in the resistant cell line.
We thank the reviewer to point out this important issue concerning DSB repair. Indeed, the negative control of this experiment was non-irradiated either radiosensitive or radioresistant PDTs. However, we did not analyze the expression of gamma-H2AX at earlier time points. We agree with the reviewer that it could be that there were less DSBs generated in the resistant PDTs.
Now we have made modification in the third paragraph of the section 3.1. Identification of different dose-response of lung cancer PDTs to X-ray based radiation therapy “Notably, g-H2AX foci were stained inside of the nucleus upon irradiation only in PDTs formed from patient 21T117, but not in those formed from 21T324 indicating that less DNA lesions made by irradiation were detected 24-hr post-irradiation in patient 21T324 than in patient 21T117 (Figure 2A).”
- Figure 2: Quantitative data of gamma H2Ax the resistant line, 21T324, is important for comparison
We agree with the reviewer that it is important to compare the gamma H2AX expression in radioresistant and radiosensitive PDTs. As shown in Figure 2A, almost no gamma H2AX expression was observed in 21T324 PDTs which could not permit us to do more quantitative data analysis.
- Line 280: “even more efficient treatment with cisplatin and vinorelbine chemotherapy for 21T080” but there was no synergistic effect observed in figure 3D
We thank the reviewer for critical reading. The conclusion “even more efficient treatment with cisplatin and vinorelbine chemotherapy for 21T080” was made based on the IC50 of cisplatin treatment alone (IC50 : 13,20, Figure 3A) and IC50 of cisplatin/vinorelbine combined treatment (IC50 : 11µM, line 246).
- How was the effect of single Vinorelbine treatment of the PDT models? Were there synergistic/ additive/antagonistic mechanisms of the 2 agents?
We have performed the single Vinorelbine treatment of the PDT models, very few effect was observed in our PDTs model under 20 µM. However, we did observe a synergistic effect of the combined cisplatin/vinorelbine treatment for PDTs 21T226 and 21T080, whereas an antagonistic effect for PDTs 20T222 as described in the last paragraph of the section 3.2. Identification of different dose-response of lung cancer PDTs to cisplatin based chemotherapy: “As compared to the treatment cisplatin alone, the combined treatment decreased the IC50 of PDTs formed from patients 21T226 (IC50 : 26.1µM), 21T080 (IC50 : 11µM) and 21T581 (IC50 : 6.5µM), indicating that the combined cisplatin/vinorelbine treatment provided a better efficiency (Figure 3D). Interestingly, a resistance to the combined chemotherapy was also observed in PDTs formed from the patient 20T222 (IC50 : 17,6µM), in which the treatment with cisplatin alone gave a better dose-response.”.
Please see the word attachment.

Reviewer 2 Report
The manuscript focuses on the evaluation of in vitro models mimicking molecular assessment of real world series NSCLC patients represents a technically correct manuscript able to elcuidate the role of in viteo models in the translational research. As regards, I would suggest to implement minor modifications to improve the readibility of this manuscript
- Introducton section, please, could the authors point of the heterogeneous landscape of clinically approved predictive biomakrers in NSCLC patients?
- In the table 2, I would reccomend implementing several modifications. I would suggest to define exon 19 deletions. Similarly, the identification of ROS-1 aberrant transcripts is needed to improve the quality of this manuscript.
- In the results section, please, could the authors underline if a significant variation of benefit may be observed in accordance with molecular profile?
- Could the authors consider if these models may be also applied to other biomarkers
Moderate english revision should be approached
Author Response
Reviewer 2
The manuscript focuses on the evaluation of in vitro models mimicking molecular assessment of real world series NSCLC patients represents a technically correct manuscript able to elcuidate the role of in viteo models in the translational research. As regards, I would suggest to implement minor modifications to improve the readibility of this manuscript
- Introducton section, please, could the authors point of the heterogeneous landscape of clinically approved predictive biomakrers in NSCLC patients?
We agree with the reviewer that the predictive biomarkers are important for the treatment of NSCLC patients. Indeed, we are now completing experiments performed with specific TKIs for KRAS and EGFR mutations, or PD-L1 with our PDT models for another publication. However, in this paper, we have more focused on the chemotherapy and radiation therapy, we didn’t list biomarkers in the introduction. We have added predictive biomarkers in the introduction section of this paper.
- In the table 2, I would recommend implementing several modifications. I would suggest to define exon 19 deletions. Similarly, the identification of ROS-1 aberrant transcripts is needed to improve the quality of this manuscript.
We agree with the reviewer that more detailed information of the mutational status could improve the quality of this manuscript. However, we have stated in this table all the results available from the clinical chart by the Strasbourg University Hospital. Given the enrolled patients were mainly early-stage lung cancer patients, NGS was not performed for all of them.
More detailed information about the mutational status will be provided and discussed in the next research paper which is in preparation.
- In the results section, please, could the authors underline if a significant variation of benefit may be observed in accordance with molecular profile?
Again, we agree with the reviewer that the variation of benefit to different treatment may be in accordance with the molecular profile. We are now preparing our next paper focusing on the targeted therapy (TKIs KRAS and EGFR) and anti-PD-L1 immunotherapy. Indeed, in this paper, we have discussed in the second and the third paragraph of the discussion section the variation of benefit of chemotherapy and radiation therapy in accordance with the KRAS mutational status. Due to limited number of patients included in this study, we cannot draw any conclusion on the variation of benefit based on molecular profile.
- Could the authors consider if these models may be also applied to other biomarkers
As indicated in the answers of the above questions, our PDT model is also applied to other biomarkers, such as targeted therapy (TKIs KRAS and EGFR) and immunotherapy (anti-PD-L1). These results will be published soon in the next research paper.
Please see the word attachment.

Round 2
Reviewer 1 Report
I would like to thank the authors for addressing the questions. Overall the response are good and the manuscript has been improved, nevertheless there are 3 points remain.
- Figure 1A: Quantitative data of cleaved-caspase 3 analyzed by IF will further strengthened conclusion
Authors's response: We agree with the reviewer that cleaved-caspase 3 analyzed by IF will further strengthened the conclusion made from cell viability assay. That’s why we performed these two analyses at the same time point (96-hr post-irradiation) to demonstrate that more apoptotic cells were detected in radiosensitive PDTs.
There may be a misunderstanding. I meant that the authors should add quantitative data of cl-caspase 3 (eg. % of positive nuclei) because only a few nuclei are shown in this figure. The quantification will show how the overall data look like. The less cells detected by viability does not necessarily reflect apoptosis (they might grow less, necrosis etc)
- Figure 2: Quantitative data of gamma H2Ax the resistant line, 21T324, is important for comparison
Authors's response: We agree with the reviewer that it is important to compare the gamma H2AX expression in radioresistant and radiosensitive PDTs. As shown in Figure 2A, almost no gamma H2AX expression was observed in 21T324 PDTs which could not permit us to do more quantitative data analysis.
Could the author put this sentence also in the manuscript "almost no gamma H2AX expression was observed in 21T324 PDTs which could not permit us to do more quantitative data analysis"
- How was the effect of single Vinorelbine treatment of the PDT models? Were there synergistic/ additive/antagonistic mechanisms of the 2 agents?
Authors's response: We have performed the single Vinorelbine treatment of the PDT models, very few effect was observed in our PDTs model under 20 µM. However, we did observe a synergistic effect of the combined cisplatin/vinorelbine treatment for PDTs 21T226 and 21T080, whereas an antagonistic effect for PDTs 20T222 as described in the last paragraph of the section 3.2. Identification of different dose-response of lung cancer PDTs to cisplatin based chemotherapy: “As compared to the treatment cisplatin alone, the combined treatment decreased the IC50 of PDTs formed from patients 21T226 (IC50 : 26.1µM), 21T080 (IC50 : 11µM) and 21T581 (IC50 : 6.5µM), indicating that the combined cisplatin/vinorelbine treatment provided a better efficiency (Figure 3D). Interestingly, a resistance to the combined chemotherapy was also observed in PDTs formed from the patient 20T222 (IC50 : 17,6µM), in which the treatment with cisplatin alone gave a better dose-response.”.
Although I think the effect of single treatment with Vinorelbine is important to include into the figure. But as the authors saw no effect of Vinorelbine at this concentration it is fine but please also mention in the manuscript that "very few effect was observed in our PDTs model under 20 µM" . This would be useful for reader for the better understanding of the data.
Author Response
Reviewer 1
I would like to thank the authors for addressing the questions. Overall the response are good and the manuscript has been improved, nevertheless there are 3 points remain.
- Figure 1A: Quantitative data of cleaved-caspase 3 analyzed by IF will further strengthened conclusion
Authors's response: We agree with the reviewer that cleaved-caspase 3 analyzed by IF will further strengthened the conclusion made from cell viability assay. That’s why we performed these two analyses at the same time point (96-hr post-irradiation) to demonstrate that more apoptotic cells were detected in radiosensitive PDTs.
There may be a misunderstanding. I meant that the authors should add quantitative data of cl-caspase 3 (eg. % of positive nuclei) because only a few nuclei are shown in this figure. The quantification will show how the overall data look like. The less cells detected by viability does not necessarily reflect apoptosis (they might grow less, necrosis etc)
We are sorry for the misunderstanding, and we agree with the reviewer that the less cells detected by viability does not necessarily reflect apoptosis, they might grow less or necrosis, etc. That’s the reason we wrote in the line 189-193 that the cleaved caspase 3 expression just demonstrated the presence of apoptotic cells in radiosensitive PDTs, while no apoptotic cells were detected in radioresistant PDTs where there was no cell death observed in viability assay. The important message that we want to give is that the X-ray irradiation could indeed induce cell apoptosis in radiosensitive PDTs, but not in radioresistant PDTs.
Figure 2: Quantitative data of gamma H2Ax the resistant line, 21T324, is important for comparison
Authors's response: We agree with the reviewer that it is important to compare the gamma H2AX expression in radioresistant and radiosensitive PDTs. As shown in Figure 2A, almost no gamma H2AX expression was observed in 21T324 PDTs which could not permit us to do more quantitative data analysis.
Could the author put this sentence also in the manuscript "almost no gamma H2AX expression was observed in 21T324 PDTs which could not permit us to do more quantitative data analysis"
We have added the following sentences in the first paragraph of the Discussion: line 298-301 “In the contrary, DNA DSBs could not be well detected in radioresistant PDTs which did not permit us to do quantitative analyses, consequently, few cell death were found 96-hr post-irradiation (Figures 1 and 2).”
- How was the effect of single Vinorelbine treatment of the PDT models? Were there synergistic/ additive/antagonistic mechanisms of the 2 agents?
Authors's response: We have performed the single Vinorelbine treatment of the PDT models, very few effect was observed in our PDTs model under 20 µM. However, we did observe a synergistic effect of the combined cisplatin/vinorelbine treatment for PDTs 21T226 and 21T080, whereas an antagonistic effect for PDTs 20T222 as described in the last paragraph of the section 3.2. Identification of different dose-response of lung cancer PDTs to cisplatin based chemotherapy: “As compared to the treatment cisplatin alone, the combined treatment decreased the IC50 of PDTs formed from patients 21T226 (IC50 : 26.1µM), 21T080 (IC50 : 11µM) and 21T581 (IC50 : 6.5µM), indicating that the combined cisplatin/vinorelbine treatment provided a better efficiency (Figure 3D). Interestingly, a resistance to the combined chemotherapy was also observed in PDTs formed from the patient 20T222 (IC50 : 17,6µM), in which the treatment with cisplatin alone gave a better dose-response.”.
Although I think the effect of single treatment with Vinorelbine is important to include into the figure. But as the authors saw no effect of Vinorelbine at this concentration it is fine but please also mention in the manuscript that "very few effect was observed in our PDTs model under 20 µM" . This would be useful for reader for the better understanding of the data.
We have added the following sentence in line 249-253: “No significant effect was observed with vinorelbine alone (10µM), however, as compared to the treatment cisplatin alone, the combined treatment decreased the IC50 of PDTs formed from patients 21T226 (IC50 : 26.1µM), 21T080 (IC50 : 11µM) and 21T581 (IC50 : 6.5µM), indicating that the combined cisplatin/vinorelbine treatment provided a better efficiency (Figure 3D).”
